# Optimal Neural Codes for Control and Estimation

**Alex Susemihl[1], Manfred Opper**
Methods of Artificial Intelligence
Technische Universität Berlin
[1] Current affiliation: Google

**Ron Meir**
Department of Electrical Engineering
Technion - Haifa

## Abstract

Agents acting in the natural world aim at selecting appropriate actions based on noisy and partial sensory observations. Many behaviors leading to decision making and action selection in a closed loop setting are naturally phrased within a control theoretic framework. Within the framework of optimal Control Theory, one is usually given a cost function which is minimized by selecting a control law based on the observations. While in standard control settings the sensors are assumed fixed, biological systems often gain from the extra flexibility of optimizing the sensors themselves. However, this sensory adaptation is geared towards control rather than perception, as is often assumed. In this work we show that sensory adaptation for control differs from sensory adaptation for perception, even for simple control setups. This implies, consistently with recent experimental results, that when studying sensory adaptation, it is essential to account for the task being performed.

## 1  Introduction

Biological systems face the difficult task of devising effective control strategies based on partial information communicated between sensors and actuators across multiple distributed networks. While the theory of Optimal Control (OC) has become widely used as a framework for studying motor control, the standard framework of OC neglects many essential attributes of biological control [1, 2, 3]. The classic formulation of closed loop OC considers a dynamical system (plant) observed through sensors which transmit their output to a controller, which in turn selects a control law that drives actuators to steer the plant. This standard view, however, ignores the fact that sensors, controllers and actuators are often distributed across multiple sub-systems, and disregards the communication channels between these sub-systems. While the importance of jointly considering control and communication within a unified framework was already clear to the pioneers of the field of Cybernetics (e.g., Wiener and Ashby), it is only in recent years that increasing effort is being devoted to the formulation of a rigorous systems-theoretic framework for control and communication (e.g., [4]). Since the ultimate objective of an agent is to select appropriate actions, it is clear that sensation and communication must subserve effective control, and should be gauged by their contribution to action selection. In fact, given the communication constraints that plague biological systems (and many current distributed systems, e.g., cellular networks, sensor arrays, power grids, etc.), a major concern of a control design is the optimization of sensory information gathering and communication (consistently with theories of active perception). For example, recent theoretical work demonstrated a sharp communication bandwidth threshold below which control (or even stabilization) cannot be achieved (for a summary of such results see [4]). Moreover, when informational constraints exists within a control setting, even simple (linear and Gaussian) problems become nonlinear and intractable, as exemplified in the famous Witsenhausen counter-example [5].

The inter-dependence between sensation, communication and control is often overlooked both in control theory and in computational neuroscience, where one assumes that the overall solution to the control problem consists of first estimating the state of the controlled system (without reference

to the control task), followed by constructing a controller based on the estimated state. This idea, referred to as the *separation principle* in Control Theory, while optimal in certain restricted settings (e.g., Linear Quadratic Gaussian (LQG) control) is, in general, sub-optimal [6]. Unfortunately, it is in general very difficult to provide optimal solutions in cases where separation fails. A special case of the separation principle, referred to as *Certainty Equivalence* (CE), occurs when the controller treats the estimated state as the true state, and forms a controller assuming full state information. It is generally overlooked, however, that although the optimal control policy does not depend directly on the observation model at hand, the expected future costs do depend on the specifics of that model [7]. In this sense, even when CE holds, costs still arise from uncertain estimates of the state and one can optimise the sensory observation model to minimise these costs, leading to sensory adaptation. At first glance, it might seem that the observation model that will minimise the expected future cost will be the observation model that minimises the estimation error. We will show, however, that this is not generally the case.

A great deal of the work in computational neuroscience has dealt independently with the problem of sensory adaptation and control, while, as stated above, these two issues are part and parcel of the same problem. In fact, it is becoming increasingly clear that biological sensory adaptation is task-dependent [8, 9]. For example, [9] demonstrates that task-dependent sensory adaptation takes place in purely motor tasks, explaining after-effect phenomena seen in experiments. In [10], the authors show that specific changes occur in sensory regions, implying sensory plasticity in motor learning. In this work we consider a simple setting for control based on spike time sensory coding, and study the optimal coding of sensory information required in order to perform a well-defined motor task. We show that even if CE holds, the optimal encoder strategy, minimising the control cost, differs from the optimal encoder required for state estimation. This result demonstrates, consistently with experiments, that neural encoding must be tailored to the task at hand. In other words, when analyzing sensory neural data, one must pay careful care to the task being performed. Interestingly, work within the distributed control community dealing with optimal assignment and selection of sensors, leads to similar conclusions and to specific schemes for sensory adaptation.

The interplay between information theory and optimal control is a central pillar of modern control theory, and we believe it must be accounted for in the computational neuroscience community. Though statistical estimation theory has become central in neural coding issues, often through the Cramér-Rao bound, there have been few studies bridging the gap between partially observed control and neural coding. We hope to narrow this gap by presenting a simple example where control and estimation yield different conclusions. The remainder of the paper is organised as follows: In section 1.1 we introduce the notation and concepts; In section 2 we derive expressions for the cost-to-go of a linear-quadratic control system observed through spikes from a dense populations of neurons; in section 3 we present the results and compare optimal codes for control and estimation with point-process filtering, Kalman filtering and LQG control; in section 4 we discuss the results and their implications.

## 1.1 Optimal Codes for Estimation and Control

We will deal throughout this paper with a dynamic system with state $X_t$, observed through noisy sensory observations $Z_t$, whose conditional distribution can be parametrised by a set of parameters $\varphi$, e.g., the widths and locations of the tuning curves of a population of neurons or the noise properties of the observation process. The conditional distribution is then given by $P_\varphi(Z_t | X_t = x)$. $Z_t$ could stand for a diffusion process dependent on $X_t$ (denoted $Y_t$) or a set of doubly-stochastic Poisson processes dependent on $X_t$ (denoted $N_t^m$). In that sense, the optimal Bayesian encoder for an estimation problem, based on the Mean Squared Error (MSE) criterion, can be written as

$$\varphi_e^* = \operatorname*{argmin}_{\varphi} \boldsymbol{E}_z \left[ \boldsymbol{E}_{X_t} \left[ \left( X_t - \hat{X}_t(Z_t) \right)^2 \Big| Z_t = z \right] \right],$$

where $\hat{X}_t(Z_t) = \boldsymbol{E}[X_t | Z_t]$ is the posterior mean, computable, in the linear Gaussian case, by the Kalman filter. We will throughout this paper consider the MMSE in the equilibrium, that is, the error in estimating $X_t$ from long sequences of observations $Z_{[0,t]}$. Similarly, considering a control problem with a cost given by

$$C(\boldsymbol{X}_0, \boldsymbol{U}_0) = \int_0^T c(X_s, U_s, s) ds + c_T(X_T),$$

where $\boldsymbol{X}_t = \{X_s | s \in [t,T]\}, \boldsymbol{U}_t = \{U_s | s \in [t,T]\}$, and so forth. We can define

$$\varphi_c^* = \operatorname*{argmin}_\varphi \boldsymbol{E}_z \min_{\boldsymbol{U}_t} \left[ \boldsymbol{E}_{\boldsymbol{X}_t} \left[ C(\boldsymbol{X}_0, \boldsymbol{U}_0) | \boldsymbol{Z}_t = z \right] \right].$$

The certainty equivalence principle states that given a control policy $\gamma^* : \mathcal{X} \to \mathcal{U}$ which minimises the cost $C$,

$$\gamma^* = \operatorname*{argmin}_\gamma C(\boldsymbol{X}_0, \boldsymbol{\gamma}(\boldsymbol{X}_0)),$$

the optimal control policy for the partially observed problem given by noisy observations $\boldsymbol{Z}_0$ of $\boldsymbol{X}_0$ is given by

$$\gamma_{CE}(\boldsymbol{Z}_t) = \gamma^* \left( \boldsymbol{E} \left[ \boldsymbol{X}_0 | \boldsymbol{Z}_t \right] \right).$$

Note that we have used the notation $\boldsymbol{\gamma}(\boldsymbol{X}_0) = \{ \gamma(X_s), s \in [0,T] \}$.

## 2 Stochastic Optimal Control

In stochastic optimal control we seek to minimize the expected future cost incurred by a system with respect to a control variable applied to that system. We will consider linear stochastic systems governed by the SDE

$$dX_t = (AX_t + BU_t)\, dt + D^{1/2} dW_t, \tag{1a}$$

with a cost given by

$$C(\boldsymbol{X}_t, \boldsymbol{U}_t, t) = \int_t^T \left( X_s^\top Q X_s + U_s^\top R U_s \right) ds + X_T^\top Q_T X_T. \tag{1b}$$

From Bellman's optimality principle or variational analysis [11], it is well known that the optimal control is given by $U_t^* = -R^{-1} B^\top S_t X_t$, where $S_t$ is the solution of the Riccati equation

$$-\dot{S}_t = Q + AS_t + S_t A^\top - S_t B^\top R^{-1} B S_t, \tag{2}$$

with boundary condition $S_T = Q_T$. The expected future cost at time $t$ and state $x$ under the optimal control is then given by

$$J(x,t) = \min_{\boldsymbol{U}_t} \boldsymbol{E} \left[ C(\boldsymbol{X}_t, \boldsymbol{U}_t, t) | X_t = x \right] = \frac{1}{2} x^\top S_t x + \int_t^T \operatorname{Tr}(DS_s)\, ds.$$

This is usually called the optimal cost-to-go. However, the system's state is not always directly accessible and we are often left with noisy observations of it. For a class of systems e.g. LQG control, CE holds and the optimal control policy for the indirectly observed control problem is simply the optimal control policy for the original control problem applied to the Bayesian estimate of the system's state. In that sense, if the CE were to hold for the system above observed through noisy observations $Y_t$ of the state at time $t$, the optimal control would be given simply by the observation-dependent control $U_t^* = -R^{-1} B^\top S_t \boldsymbol{E} \left[ X_t | Y_t \right]$ [7].

Though CE, when applicable, gives us a simple way to determine the optimal control, when considering neural systems we are often interested in finding the optimal encoder, or the optimal observation model for a given system. That is equivalent to finding the optimal tuning function for a given neuron model. Since CE neatly separates the estimation and control steps, it would be tempting to assume the optimal codes obtained for an estimation problem would also be optimal for an associated control problem. We will show here that this is not the case.

As an illustration, let us consider the case of LQG with incomplete state information. One could, for example, take the observations to be a secondary process $Y_t$, which itself is a solution to

$$dY_t = FX_t dt + G^{1/2} dV_t,$$

the optimal cost-to-go would then be given by [11]

$$J(y,t) = \min_{\boldsymbol{U}_t} \boldsymbol{E} \left[ C(\boldsymbol{X}_t, \boldsymbol{U}_t, t) \big| Y_{[0,t]} = y \right] \tag{3}$$

$$= \nu_t^\top S_t \nu_t + \operatorname{Tr}(K_t S_t) + \int_t^T \operatorname{Tr}(DS_s)\, ds + \int_t^T \operatorname{Tr}\left( S_s BR^{-1} B^\top S_s K_s \right) ds,$$

where we have defined $Y_{[0,t]} = \{Y_s, s \in [0,t]\}$, $\nu_t = \mathbf{E}[X_t|Y_{[0,t]}]$ and $K_t = \text{cov}[X_t|Y_{[0,t]}]$. We give a demonstration of these results in the SI, but for a thorough review see [11]. Note that through the last term in equation (3) the cost-to-go now depends on the parameters of the $Y_t$ process. More precisely, the variance of the distribution of $X_s$ given $Y_t$, for $s > t$ obeys the ODE

$$\dot{K}_t = AK_t + K_tA^\top + D - K_tF^\top G^{-1}FK_t. \tag{4}$$

One could then choose the matrices $F$ and $G$ in such a way as to minimise the contribution of the rightmost term in equation (3). Note that in the LQG case this is not particularly interesting, as the conclusion is simply that we should strive to make $K_t$ as small as possible, by making the term $F^\top G^{-1}F$ as large as possible. This translates to choosing an observation process with very strong steering from the unobserved process (large $F$) and a very small noise (small $G$). One case that provides some more interesting situations is if we consider a two-dimensional system, where we are restricted to a noise covariance with constant determinant. That means the hypervolume spanned by the eigenvectors of the covariance matrix is constant. We will compare this case with the Poisson-coded case below.

## 2.1 LQG Control with Dense Gauss-Poisson Codes

Let us now consider the case of the system given by equation (1a), but instead of observing the system directly we observe a set of doubly-stochastic Poisson processes $\{N_t^m\}$ with rates given by

$$\lambda^m(x) = \phi \exp\left[-\frac{1}{2}(x - \theta_m)^\top P^\dagger (x - \theta_m)\right]. \tag{5}$$

To clarify, the process $N_t^m$ is a counting process which counts how many spikes the neuron $m$ has fired up to time $t$. In that sense, the differential of the counting process $dN_t^m$ will give the spike train process, a sum of Dirac delta functions placed at the times of spikes fired by neuron $m$. Here $P^\dagger$ denotes the pseudo-inverse of $P$, which is used to allow for tuning functions that do not depend on certain coordinates of the stimulus $x$. Furthermore, we will assume that the tuning centre $\theta_m$ are such that the probability of observing a spike of any neuron at a given time $\hat{\lambda} = \sum_m \lambda^m(x)$ is independent of the specific value of the world state $x$. This can be a consequence of either a dense packing of the tuning centres $\theta_m$ along a given dimension of $x$, or of an absolute insensitivity to that aspect of $x$ through a null element in the diagonal of $P^\dagger$. This is often called the dense coding hypothesis [12]. It can be readily be shown that the filtering distribution is given by $P(X_t|\{N_{[0,t)}\}) = \mathcal{N}(\mu_t, \Sigma_t)$, where the mean and covariance are solutions to the stochastic differential equations (see [13])

$$d\mu_t = (A\mu_t + BU_t)\,dt + \sum_m \Sigma_t \left(I + P^\dagger \Sigma_t\right)^{-1} P^\dagger (\theta_m - \mu_t)\,dN_t^m, \tag{6a}$$

$$d\Sigma_t = \left(A\Sigma_t + \Sigma_t A^\top + D\right)dt - \Sigma_t P^\dagger \Sigma_t \left(I + P^\dagger \Sigma_t\right)^{-1} dN_t, \tag{6b}$$

where we have defined $\mu_t = \mathbf{E}[X_t|\{N_{[0,t]}^m\}]$ and $\Sigma_t = \text{cov}[X_t|\{N_{[0,t]}^m\}]$. Note that we have also defined $N_{[0,t]}^m = \{N_s^m|s \in [0,t]\}$, the history of the process $N_s^m$ up to time $t$, and $N_t = \sum_m N_t^m$. Using Lemma 7.1 from [11] provides a simple connection between the cost function and the solution of the associated Ricatti equation for a stochastic process. We have

$$C(\boldsymbol{X}_t, \boldsymbol{U}_t, t) = X_T^\top Q_T X_T + \int_t^T \left[X_s^\top Q X_s + U_s^\top R U_s\right] ds$$

$$= X_t^\top S_t X_t + \int_t^T (U_s + R^{-1}B^\top S_s X_s)^\top R(U_s + R^{-1}B^\top S_s X_s) ds$$

$$+ \int_t^T \text{Tr}(DS_s)ds + \int_t^T dW_s^\top D^{\top/2} S_s X_s ds + \int_t^T X_s^\top S_s D^{1/2} dW_s.$$

We can average over $P(\boldsymbol{X}_t, \boldsymbol{N}_t|\{N_{[0,t)}\})$ to obtain the expected future cost. That gives us

$$\mu_t^\top S_t \mu_t + \text{Tr}(\Sigma_t S_t) + \mathbf{E}\left[\int_t^T (U_s + R^{-1}B^\top S_s X_s)^\top R(U_s + R^{-1}B^\top S_s X_s)ds \,\middle|\, \{N_{[0,t)}\}\right] + \int_t^T \text{Tr}(DS_s)ds$$

We can evaluate the average over $P(\boldsymbol{X}_t, \{\boldsymbol{N}_t^m\}|\{N_{[0,t)}^m\})$ in two steps, by first averaging over the Gaussian densities $P(X_s|\{N_{[0,s]}^m\})$ and then over $P(\{N_{[0,s]}\}|\{N_{[0,t)}\})$. The average gives

$$\boldsymbol{E}\left[\left.\int_t^T (U_s + R^{-1}B^\top S_s\mu_s)^\top R(U_s + R^{-1}B^\top S_s\mu_s) + \operatorname{Tr}\left[S_s BR^{-1}B^\top S_s \Sigma_s(\{N_{[0,s]}\})\right] ds\right| \{N_{[0,t)}\}\right],$$

where $\mu_s$ and $\Sigma_s$ are the mean and variance associated with the distribution $P(X_s|\{N_{[0,s)}\})$. Note that choosing $U_s = -R^{-1}B^\top S_s\mu_s$ will minimise the expression above, consistently with CE. The optimal cost-to-go is therefore given by

$$
\begin{aligned}
J(\{N_{[0,t)}\}, t) =& \mu_t^\top S_t \mu_t + \operatorname{Tr}(\Sigma_t S_t) \\
&+ \int_t^T \operatorname{Tr}(DS_s)\, ds + \int_t^T \operatorname{Tr}\left(S_s BR^{-1}B^\top S_s \boldsymbol{E}\left[\Sigma_s(\{N_{[0,s]}\})|\{N_{[0,t)}\}\right]\right) ds
\end{aligned}
\tag{7}
$$

Note that the only term in the cost-to-go function that depends on the parameters of the encoders is the rightmost term and it depends on it only through the average over future paths of the filtering variance $\Sigma_s$. The average of the future covariance matrix is precisely the MMSE for the filtering problem conditioned on the belief state at time $t$ [13]. We can therefore analyse the quality of an encoder for a control task by looking at the values of the term on the right for different encoding parameters. Furthermore, since the dynamics of $\Sigma_t$ given by equation (6b) is Markovian, we can write the average $\boldsymbol{E}\left[\Sigma_s|\{N_{[0,t)}\}\right]$ as $\boldsymbol{E}\left[\Sigma_s|\Sigma_t\right]$. We will define then the function $f(\Sigma, t)$ which gives us the uncertainty-related expected future cost for the control problem as

$$f(\Sigma, t) = \int_t^T \operatorname{Tr}\left(S_s BR^{-1}B^\top S_s \boldsymbol{E}\left[\Sigma_s|\Sigma_t = \Sigma\right]\right) ds. \tag{8}$$

## 2.2 Mutual Information

Many results in information theory are formulated in terms of the mutual information of the communication channel $P_\varphi(Y|X)$. For example, the maximum cost reduction achievable with $R$ bits of information about an unobserved variable $X$ has been shown to be a function of the rate-distortion function with the cost as the distortion function [14]. More recently there has also been a lot of interest in the so-called I-MMSE relations, which provide connections between the mutual information of a channel and the minimal mean squared error of the Bayes estimator derived from the same channel [15, 16]. The mutual information for the cases we are considering is not particularly complex, as all distributions are Gaussians. Let us denote by $\Sigma_t^0$ the covariance of of the unobserved process $X_t$ conditioned on some initial Gaussian distribution $P_0 = \mathcal{N}(\mu_0, \Sigma_0)$ at time 0. We can then consider the Mutual Information between the stimulus at time $t$, $X_t$, and the observations up to time t, $Y_{[0,t]}$ or $N_{[0,t]}$. For the LQG/Kalman case we have simply

$$I(X_t; Y_{[0,t]}|P_0) = \int dx\, dy P(x,y)\left[\log P(x|y) - \log P(x)\right] = \log|\Sigma_t^0| - \log|\Sigma_t|,$$

where $\Sigma_t$ is a solution of equation (4). For the Dense Gauss-Poisson code, we can also write

$$I(X_t; N_t|P_0) = \int dx\, dn\, P(x,n)\left[\log P(x|n) - \log P(x)\right] = \log|\Sigma_t^0| - \boldsymbol{E}_{N_{[0,t]}}\left[\log|\Sigma_t(N_{[0,t]})|\right],$$

where $\Sigma_t(N_{[0,t]})$ is a solution to the stochastic differential equation (6b) for the given value of $N_{[0,t]}$.

## 3 Optimal Neural Codes for Estimation and Control

What could be the reasons for an optimal code for an estimation problem to be sub-optimal for a control problem? We present examples that show two possible reasons for different optimal coding strategies in estimation and control. First, one should note that control problems are often defined over a finite time horizon. One set of classical experiments involves reaching for a target under time constraints [3]. If we take the maximal firing rate of the neurons ($\phi$) to be constant while varying the width of the tuning functions, this will lead the number of observed spikes to be inversely proportional to the precision of those spikes, forcing a trade-off between the number of observations

and their quality. This trade-off can be tilted to either side in the case of control depending on the information available at the start of the problem. If we are given complete information on the system state at the initial time $0$, the encoder needs fewer spikes to reliably estimate the system's state throughout the duration of the control experiment, and the optimal encoder will be tilted towards a lower number of spikes with higher precision. Conversely, if at the beginning of the experiment we have very little information about the system's state, reflected in a very broad distribution, the encoder will be forced towards lower precision spikes with higher frequency. These results are discussed in section 3.1.

Secondly, one should note that the optimal encoder for estimation does not take into account the differential weighting of different dimensions of the system's state. When considering a multidimensional estimation problem, the optimal encoder will generally allocate all its resources equally between the dimensions of the system's state. In the framework presented we can think of the dimensions as the singular vectors of the tuning matrix $P$ and the resources allocated to it are the singular values. In this sense, we will consider a set of coding strategies defined by matrices $P$ of constant determinant in section 3.2. This constrains the overall firing rate of the population of neurons to be constant, and we can then consider how the population will best allocate its observations between these dimensions. Clearly, if we have an anisotropic control problem, which places a higher importance in controlling one dimension, the optimal encoder for the control problem will be expected to allocate more resources to that dimension. This is indeed shown to be the case for the Poisson codes considered, as well as for a simple LQG problem when we constrain the noise covariance to have the same structure.

We do not mean our analysis to be exhaustive as to the factors leading to different optimal codes in estimation and control settings, as the general problem is intractable, and indeed, is not even separable. We intend this to be a proof of concept showing two cases in which the analogy between control and estimation breaks down.

## 3.1 The Trade-off Between Precision and Frequency of Observations

In this section we consider populations of neurons with tuning functions as given by equation (5) with tuning centers $\theta_m$ distributed along a one- dimensional line. In the case of the Ornstein-Uhlenbeck process these will be simply one-dimensional values $\theta_m$ whereas in the case of the stochastic oscillator, we will consider tuning centres of the form $\theta_m = (\eta_m, 0)^\top$, filling only the first dimension of the stimulus space. Note that in both cases the (dense) population firing rate $\hat{\lambda} = \sum_m \lambda_m(x)$ will be given by $\hat{\lambda} = \sqrt{2\pi}p\phi/|\Delta\theta|$, where $\Delta\theta$ is the separation between neighbouring tuning centres $\theta_m$.

The Ornstein-Uhlenbeck (OU) process controlled by a process $U_t$ is given by the SDE

$$dX_t = (bU_t - \gamma X_t)dt + D^{1/2}dW_t.$$

Equation (7) can then be solved by simulating the dynamics of $\Sigma_s$. This has been considered extensively in [13] and we refer to the results therein. Specifically, it has been found that the dynamics of the average can be approximated in a mean-field approach yielding surprisingly good results. The evolution of the average posterior variance is given by the average of equation (6b), which involves nonlinear averages over the covariances. These are intractable, but a simple mean-field approach yields the approximate equation for the evolution of the average $\langle \Sigma_s \rangle = \boldsymbol{E}\left[\Sigma_s | \Sigma_0\right]$

$$\frac{d\langle \Sigma_s \rangle}{ds} = A\langle \Sigma_s \rangle + \langle \Sigma_s \rangle^\top A^\top + D - \hat{\lambda}\langle \Sigma_s \rangle P^\dagger \langle \Sigma_s \rangle \left(I + P^\dagger \langle \Sigma_s \rangle\right)^{-1}.$$

The alternative is to simulate the stochastic dynamics of $\Sigma_t$ for a large number of samples and compute numerical averages. These results can be directly employed to evaluate the optimal cost-to-go in the control problem $f(\Sigma, t)$.

Alternatively, we can look at a system with more complex dynamics, and we take as an example the stochastic damped harmonic oscillator given by the system of equations

$$\dot{X}_t = V_t, \quad dV_t = \left(bU_t - \gamma V_t - \omega^2 X_t\right)dt + \eta^{1/2}dW_t. \tag{9}$$

Furthermore, we assume that the tuning functions only depend on the position of the oscillator, therefore not giving us any information about the velocity. The controller in turn seeks to keep the

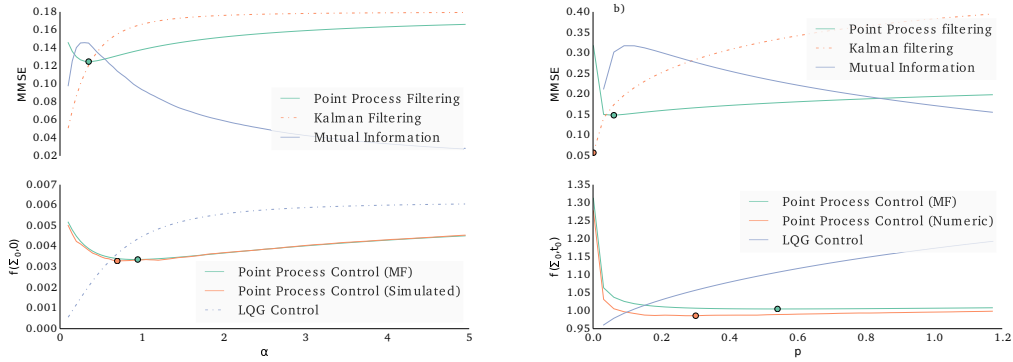

Figure 1: The trade-off between the precision and the frequency of spikes is illustrated for the OU process (a) and the stochastic oscillator (b). In both figures, the initial condition has a very uncertain estimate of the system's state, biasing the optimal tuning width towards higher values. This forces the encoder to amass the maximum number of observations within the duration of the control experiment. Parameters for figure (a) were: $T = 2$, $\gamma = 1.0$, $\eta = 0.6$, $b = 0.2$, $\phi = 0.1$, $\Delta\theta = 0.05$, $Q = 0.1$, $Q_T = 0.001$, $R = 0.1$. Parameters for figure (b) were $T = 5$, $\gamma = 0.4$, $\omega = 0.8$, $\eta = 0.4$, $r = 0.4$, $q = 0.4$, $Q_T = 0$, $\phi = 0.5$, $\Delta\theta = 0.1$.

oscillator close to the origin while steering only the velocity. This can be achieved by the choice of matrices $A = (0, 1; -\omega^2, -\gamma)$, $B = (0, 0; 0, b)$, $D = (0, 0; 0, \eta^2)$, $R = (0, 0; 0, r)$, $Q = (q, 0; 0, 0)$ and $P = (p^2, 0; 0, 0)$.

In figure 1 we provide the uncertainty-dependent costs for LQG control, for the Poisson observed control, as well as the MMSE for the Poisson filtering problem and for a Kalman-Bucy filter with the same noise covariance matrix $P$. This illustrates nicely the difference between Kalman filtering and the Gauss-Poisson filtering considered here. The Kalman filter MSE has a simple, monotonically increasing dependence on the noise covariance, and one should simply strive to design sensors with the highest possible precision ($p = 0$) to minimise the MMSE and control costs. The Poisson case leads to optimal performance at a non-zero value of $p$. Importantly the optimal values of $p$ for estimation and control differ. Furthermore, in view of section 2.2, we also plotted the mutual information between the process $X_t$ and the observation process $N_t$, to illustrate that information-based arguments would lead to the same optimal encoder as MMSE-based arguments.

### 3.2 Allocating Observation Resources in Anisotropic Control Problems

A second factor that could lead to different optimal encoders in estimation and control is the structure of the cost function $C$. Specifically, if the cost functions depends more strongly on a certain coordinate of the system's state, uncertainty in that particular coordinate will have a higher impact on expected future costs than uncertainty in other coordinates. We will here consider two simple linear control systems observed by a population of neurons restricted to a certain firing rate. This can be thought of as a metabolic constraint, since the regeneration of membrane potential necessary for action potential generation is one of the most significant metabolic expenditures for neurons [17]. This will lead to a trade-off, where an increase in precision in one coordinate will result in a decrease in precision in the other coordinate.

We consider a population of neurons whose tuning functions cover a two-dimensional space. Taking a two-dimensional isotropic OU system with state $X_t = (X_{1,t}, X_{2,t})^\top$ where both dimensions are uncoupled, we can consider a population with tuning centres $\theta_m = (\eta_1^m, \eta_2^m)^\top$ densely covering the stimulus space. To consider a smoother class of stochastic systems we will also consider a two-dimensional stochastic oscillator with state $X_t = (X_{1,t}, V_{1,t}, X_{2,t}, V_{2,t})^\top$, where again, both dimensions are uncoupled, and the tuning centres of the form $\theta_m = (\eta_1^m, 0, \eta_2^m, 0)^\top$, covering densely the position space, but not the velocity space.

Since we are interested in the case of limited resources, we will restrict ourselves to populations with a tuning matrix $P$ yielding a constant population firing rate. We can parametrise these simply as $P_{OU}(\zeta) = p^2 \operatorname{Diag}(\tan(\zeta), \cot an(\zeta))$, for the OU case and $P_{Osc}(\zeta) =$

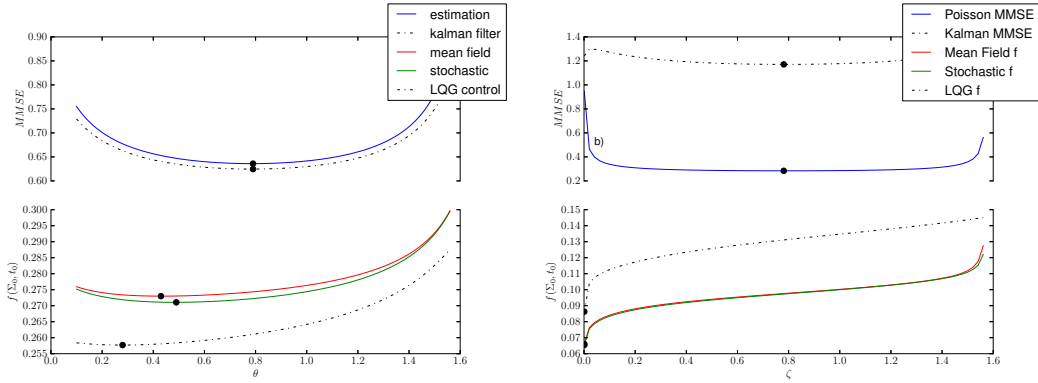

Figure 2: The differential allocation of resources in control and estimation for the OU process (left) and the stochastic oscillator (right). Even though the estimation MMSE leads to a symmetric optimal encoder both in the Poisson and in the Kalman filtering problem, the optimal encoders for the control problem are asymmetric, allocating more resources to the first coordinate of the stimulus.

$p^2 \operatorname{Diag}(\tan(\zeta), 0, \cot(\zeta), 0)$ for the stochastic oscillator, where $\zeta \in (0, \pi/2)$. Note that this will yield the firing rate $\hat{\lambda} = 2\pi p \phi/(\Delta\theta)^2$, independent of the specifics of the matrix $P$.

We can then compare the performance of all observers with the same firing rate in both control and estimation tasks. As mentioned, we are interested in control problems where the cost functions are anisotropic, that is, one dimension of the system's state vector contributes more heavily to the cost function. To study this case we consider cost functions of the type

$$c(X_t, U_t) = Q_1 X_{1,t}^2 + Q_2 X_{2,t}^2 + R_1 U_{1,t}^2 + R_2 U_{2,t}^2.$$

This again, can be readily cast into the formalism introduced above, with a suitable choice of matrices $Q$ and $R$ for both the OU process as for the stochastic oscillator. We will also consider the case where the first dimension of $X_t$ contributes more strongly to the state costs (i.e., $Q_1 > Q_2$).

The filtering error can be obtained from the formalism developed in [13] in the case of Poisson observations and directly from the Kalman-Bucy equations in the case of Kalman filtering [18]. For LQG control, one can simply solve the control problem for the system mentioned using the standard methods (see e.g. [11]). The Poisson-coded version of the control problem can be solved using either direct simulation of the dynamics of $\Sigma_s$ or by a mean-field approach which has been shown to yield excellent results for the system at hand. These results are summarised in figure 2, with similar notation to that in figure 1. Note the extreme example of the stochastic oscillator, where the optimal encoder is concentrating all the resources in one dimension, essentially ignoring the second dimension.

## 4   Conclusion and Discussion

We have here shown that the optimal encoding strategies for a partially observed control problem is not the same as the optimal encoding strategy for the associated state estimation problem. Note that this is a natural consequence of considering noise covariances with a constant determinant in the case of Kalman filtering and LQG control, but it is by no means trivial in the case of Poisson-coded processes. For a class of stochastic processes for which the certainty equivalence principle holds we have provided an exact expression for the optimal cost-to-go and have shown that minimising this cost provides us with an encoder that in fact minimises the incurred cost in the control problem.

Optimality arguments are central to many parts of computational neuroscience, but it seems that partial observability and the importance of combining adaptive state estimation and control have rarely been considered in this literature, although supported by recent experiments. We believe the present work, while treating only a small subset of the formalisms used in neuroscience, provides a first insight into the differences between estimation and control. Much emphasis has been placed on tracing the parallels between the two (see [19, 20], for example), but one must not forget to take into account the differences as well.

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
