[Reviews · NeurIPS 2014]

Submitted by Assigned_Reviewer_25

Summary
-------

The authors study stochastic optimal control problems with incomplete state information.
In particular, they consider problems where the sensors are adaptive.
They compare sensor configurations which are optimal under a standard signal detection paradigm with sensor configurations that are optimal for a given control problem.

Comments
--------

Optimal sensor adaptation has a long history in computational neuroscience. Usually, optimality is defined wrt estimation (infomax etc).
The main point of the paper, that optimal sensor adaptation for control problems will yield different results in general, is very important and could potentially have a big impact on the field.
Standard optimal coding approaches were often criticized but concrete alternatives were lacking.
Here the authors propose optimal control as a guiding principle and illustrate this with a few well-chosen examples.
Not being an expert in control theory, the paper seems technically sound (I checked most equations, not all).
It is well written, if a little too dense at times. Some results / examples require further elaboration (details below), it might be worth dropping one (eg the harmonic oscillator in 3.1) in favor of more details for the other examples.

Specific comments:

- l22: ``However, this sensory adaptation is geared towards control rather than perception, as is often assumed.'' In principle I agree with this statement. However, there's an interesting discussion to be had based on the following argument: Assume that the sensors are used in many different control problems, and therefore adapting them for a single one might result in poor performance on the others. In this situation it might be beneficial to adapt the sensory based on optimal estimation after all. I'd be curious to hear the opinion of the authors on this point.

- l103: ``MMSE'' is probably the minimum MSE, but it's never defined. Furthermore, the authors seem to use MSE / MMSE inconsistently, eg in the legend of figs 3&4, shouldn't it be just MSE instead of MMSE?

- l110: Shouldn't the minimization be over $\mathbf U_0$ instead of $\mathbf U_t$? And similarly the expectation over $\mathbf X_0$, etc...

- l117: The notation here is a bit sloppy I think. On the RHS of the eqn, it prob should be $\gamma$ applied to the time-series of filtered estimates of $X_t$ given $Z_{[0,t]}$. By definition in l108 $\mathbf Z_t$ is the future of $Z$, and not the past...

- l146: typo: $E[X_t\vert Y_t]$ should prob be $E[X_t\vert Y_{[0,t]}$

- l158, eqn 3: the authors state that only the third term depends on the parameters $F,G$ of the observation process $Y$. For $t > 0$, why don't the first two terms depend on $F,G$ via $\nu_t$ and $K_t$? For $t=0$, I agree that only the 3rd term depends on $F,G$.

- l182, eqn 5: Concerning the dense coding hypothesis: I was wondering weather it can be achieved with a finite number of neurons each having an RBF / squared-exponential receptive field (my intuition tells might not, but I might be wrong). If one indeed needs infinitely many neurons, than one could possibly replace the spike process $dN_t$ with a diffusion / Wiener process and hence simplify eqns 6a and 6b. Is this possible? The authors might be interested in the use of the dense packing hypothesis in [Implicit embedding of prior probabilities in optimally efficient neural populations. D Ganguli and E P Simoncelli. ArXiv]

- l258: typo, this should probably be the mutual information between $X_t$ and $\{N^m_{[0,t]}\}_{m=1,\ldots,M}$

- l270: ``This trade-off can be tilted to either side in the case of control depending on the information available at the start of the problem.'' I don't fully understand this point. Furthermore, I cant find the promised detailed discussion of this point in section 3.1.

- fig 2: The left and right panels use different labels in the legends for the same quantities. Left panel: the x-axis label should probably be also $\zeta$

- l413: ``Note the extreme example of the stochastic oscillator, where the optimal encoder is concentrating all the resources in one dimension, essentially ignoring the second dimension.'' It's important to spell out in detail what this means for the tuning curve width in direction and perpendicular to the ``important control dimension''

Summary: The paper makes a novel and important contribution to the optimal coding debate in computational neuroscience.

Submitted by Assigned_Reviewer_26

Paper 1562

Optimal Neural Codes for Control and Estimation

The authors describe how systems with observation model parameters optimized for control and estimation problems differ. They assert that the estimation framework is applicable to sensory neural coding and that the control framework is applicable to motor neural control. The paper describes the formalisms and then proceeds to explain how the two are different from one another with a couple simple illustrative examples. The paper is clearly written, and makes an important point: systems that are optimal in terms of estimation criteria may not be optimal in terms of different control-theoretic criteria. However, this point, while important, strikes me as not particularly surprising or novel. Indeed, it would have been more surprising if different objective functions (i.e. corresponding to different sensory or control tasks) had led to the same optimal parameters. The calculations presented to support the analyses here are fairly standard. Thus, in summary, while it’s a well-written and elegant paper, and makes an important point, I did not rank this paper very highly, since the novelty of the contribution here seems modest.

Specific points:

Section 2 gives some math for the different objectives, but this is all standard I believe (?). I would have preferred to see MMSE estimation comparisons rather than mutual information ones -- or at least a concrete statement as to how the results of [15,16] apply here -- if there isn't a succinct connection between how the optimal bayes estimator error relates to the MSE here, I would rather see MMSE results here because otherwise it isn't clear how to compare.

I'm not sure I got any useful insights out of section 3.1. It was meant to clarify a specific tradeoff, but it wasn't obvious this particular point needed clarification. Similarly, I found 3.2 overly contrived and without much conceptual benefit - this is not to say that it isn't a useful regime, just not obviously something that needed so much space. Basically sections 3 - 3.2 focused on particular consequences of comparing two problems with different objectives, but it wasn't made clear enough why these features are interesting for neuroscience or unintuitive enough to be significant (or non-obvious).

I would have benefited from a more specific formal description of how the model was being used for the neural case -- what are the constraints on the encoder, is it time-varying, can it change arbitrarily quickly, are these assumptions reasonable, when?

Summary: The authors describe how systems with observation model parameters optimized for control and estimation problems differ. While it’s a well-written and elegant paper, and makes an important point, I did not rank this paper very highly, since the novelty of the contribution here seems modest.

Submitted by Assigned_Reviewer_44

This paper provides answers to an important question: how does the structure of a control problem govern the optimal encoding of sensory inferences? The authors analyze several scenarios where the optimal sensory encoding departs from classical schemes (e.g., the Kalman filter) when the encoding is chosen to optimize the cost-to-go.

These theoretical results may have important consequences for our understanding of brain function; for example, it could potentially explain why early sensory representations are influenced by rewards. However, I found that the writing style was dense and opaque, providing insufficient motivation and intuitions. (This is at least true for the average computational neuroscientist; a signal processing expert might find it easier going).

I also think that the authors could do a better job connecting to real problems, either in neuroscience or control theory. The problems they analyze are very abstract, divorced from any real application. I think exploring such applications would help readers better appreciate these theoretical results.

Minor comments:
- line 361: "cost functions depends" -> "cost function depends"
Summary: An interesting and potentially significant analysis of sensory representation in control problems. The exposition is dense and opaque, needing more intuitive explanations and connections to real problems, thus limiting its broader appeal.
Author Feedback
Author rebuttal: All reviewers explicitly stated that this was an "important" topic, but some have questioned its novelty. We do not believe the presented results to have
been presented elsewhere, neither in the context of computational neuroscience nor in control theory. Most results relating estimation and control focus on the duality relation between them, with no mention of the issue of optimal encoding. While there is an expanding literature in Control Theory related to network control, where communication limitations play an important role (review in [4]), we are not aware of any studies in that context dealing with adaptive sensory processing for control. In standard control settings the sensors are assumed fixed, though this notion is changing within the field of network control. This literature has, to the best of our knowledge, not been discussed in the context of neural computation (e.g., bell-shaped tuning curves, point process observations), which we believe (and all reviewers seem to concur) is an important and timely topic.

Rev 25

l. 22: This is a very interesting discussion, and could be a natural extension to this work. Though one way to adapt to different control problems using the same encoder would be to fall back on the MMSE, this need not be optimal generally. One example is the distribution of photoreceptors in the retina, which are used in many behavioural tasks. Their distribution is not MMSE-optimal, however, favouring central regions, which are behaviourally more relevant.

l.103, l.110,117,146,258 These will be corrected.

l. 158 The mean and covariance of the Gaussian belief state at time t are a function of the observations up to time t, and don't directly depend on the value of F and G. The dynamics of the observation process itself depends on F and G, but conditioned on the observation process, K_t and \nu_t are independent of F and G.

l.182 The full filtering equations for the problem are given in (Snyder, D, 1972 Filtering...). It suffices for the Gaussian approximation to be exact that the product of P(x)(\hat{\lambda}-\lambda(x)) is vanishingly small. Thus we do not need infinite neurons, it suffices to cover densely the region where the stimulus has a reasonable probability of occurring. A diffusion approximation wouldn't be appropriate in the limit of infinite neurons, but only in the limit of very broadly tuned neurons, or of very frequent spikes. Note that the limit of infinite neurons does not imply arbitrarily dense packing.

l.270 The trade-off has only been demonstrated towards one side (a very uncertain initial belief state shifts the optimal encoder towards broader functions, as shown in figure 1). This should have been made explicit.

l.413 A very small tuning width in one direction would lead to a very broad tuning width in the other, leading the encoder to be nearly independent of one of the coordinates of the stimulus. This would be equivalent to an array of sensors detecting a two-dimensional stimulus to ignore the x-coordinate in favour of the more costly y-coordinate. This will be clarified in the paper.

Rev 26

The reviewer states that these calculations are fairly standard, yet we have not found such calculations anywhere. We would be very thankful if the reviewer directed us to a reference.

Section 2: The presentation of the LQG problem with Gauss-Poisson observations hasn't been presented elsewhere to the best of our knowledge, and the discussion of uncertainty costs for optimal encoding are novel in the context of sensor adaptation/design too. As for the discussion of the MI, this might be misplaced, and could be more focused on the MMSE.

Section 3: This section aims to explain two simple effects in two different models. The discussion is somewhat dense, and could possibly be rephrased to be more easily read. The trade-off discussed in section 3.1 is present in estimation as well and has been discussed elsewhere. The main point of the paper is that while the trade-off is present in a control problem as well, the optimal encoder is still different from the estimation problem. As for section 3.2, we sought to illustrate a simple case where a neural population is trying to code for two variables jointly on a limited energy budget. The first paragraph of section 3.2 motivates this setup, but could be made clearer.

Rev 44

A lot of recent has demonstrated the modification of sensory responses due to motor activity and learning. The neuroscience references provided on this topic were somewhat non-representative ([10] is the most relevant). Much of this work comes from the laboratory of Prof. Ostry at McGill, and many pertinent references can be found there (specifically, the 2011 paper "Sensory change following motor learning'' by Mattar et al, provides a brief history and references of the effect). This problem is novel in the context of control theory as well, so there are no "canonical" models to relate to. We have preferred to stay within the realm of exactly solvable problems to gain deeper intuition. Control problems are generally intractable, and approximate solutions have the disadvantage of clouding the conclusions. We are aware of only a very limited number of computational studies on these issues. A NIPS 2008 paper by Haith et al. provided experimental results related to adaptation in hand movement experiments and a highly simplified linear control setup (scalar linear dynamics and linear observation), with unit state and observation matrices. A 2013 paper by Ito et al., "Computational model of motor learning and perceptual change", proposed and simulated a specific arm model and (non-optimal) control strategy. Our work differs in being framed within the theory of Optimal Control, considering more general dynamic settings and neurally plausible observation models, and providing a simple intuitive and analytically tractable model and framework. Both these works will be mentioned in the final paper.